# Hyperspectral Imaging and Chemometrics for Authentication of Extra Virgin Olive Oil: A Comparative Approach with FTIR, UV-VIS, Raman, and GC-MS

**DOI:** 10.3390/foods12030429

**Published:** 2023-01-17

**Authors:** Derick Malavi, Amin Nikkhah, Katleen Raes, Sam Van Haute

**Affiliations:** 1Department of Food Technology, Safety and Health, Faculty of Bioscience Engineering, Ghent University, Coupure Links 653, 9000 Ghent, Belgium; 2Center for Food Chemistry and Technology, Ghent University Global Campus, Incheon 21985, Republic of Korea; 3Friedman School of Nutrition Science and Policy, Tufts University, Boston, MA 02111, USA

**Keywords:** hyperspectral imaging, extra virgin olive oil, adulteration, authenticity, edible oils

## Abstract

Limited information on monitoring adulteration in extra virgin olive oil (EVOO) by hyperspectral imaging (HSI) exists. This work presents a comparative study of chemometrics for the authentication and quantification of adulteration in EVOO with cheaper edible oils using GC-MS, HSI, FTIR, Raman and UV-Vis spectroscopies. The adulteration mixtures were prepared by separately blending safflower oil, corn oil, soybean oil, canola oil, sunflower oil, and sesame oil with authentic EVOO in different concentrations (0–20%, m/m). Partial least squares-discriminant analysis (PLS-DA) and PLS regression models were then built for the classification and quantification of adulteration in olive oil, respectively. HSI, FTIR, UV-Vis, Raman, and GC-MS combined with PLS-DA achieved correct classification accuracies of 100%, 99.8%, 99.6%, 96.6%, and 93.7%, respectively, in the discrimination of authentic and adulterated olive oil. The overall PLS regression model using HSI data was the best in predicting the concentration of adulterants in olive oil with a low root mean square error of prediction (RMSEP) of 1.1%, high R^2^_pred_ (0.97), and high residual predictive deviation (RPD) of 6.0. The findings suggest the potential of HSI technology as a fast and non-destructive technique to control fraud in the olive oil industry.

## 1. Introduction

The International Olive Oil Council (IOC) defines virgin olive oil as the product from the fruit of the olive trees (*Olea europaea* L.) exclusively obtained by mechanical or other physical methods that do not alter its physical, chemical, or organoleptic properties [1]. The IOC has designated extra virgin olive oil (EVOO) as the highest grade of olive oil in the market.

EVOO is an important part of the Mediterranean diet. It is highly appreciated worldwide due to its nutritional, healthy, and unique organoleptic attributes. It contains several health-promoting compounds such as monounsaturated fatty acids (oleic acid), tocopherols, carotenoids, and phenolic compounds. Some of these compounds possess significant biological activities that prevent coronary heart diseases and fight specific cancer cells [2]. EVOO is priced 3–5 times higher than common edible oils due to its high nutritional value. The unique nutritional value, high demand and high cost of production make EVOO susceptible to economically motivated adulteration.

EVOO is adulterated in diverse ways such as accidental contamination during oil processing, addition of cheaper edible oils and mislabeling of less expensive olive oil categories. Partial or total replacement of olive oil with other edible oils such as corn, soybean, sunflower, olive pomace, walnut and rapeseed oils has been the most frequent form of adulteration in the market [3,4,5]. Other than misleading the consumer, adulteration of EVOO can create safety issues related to allergens [6].

Consequently, authentication and control of adulteration are crucial for the olive oil industry to ensure olive oil quality and fair trade, as well as to safeguard the health and safety of the consumer. The European Commission, International Olive Council and Codex Committee on Fats and Oils are mandated in anti-fraud regulations of EVOO. They have established olive oil regulations based on the country of origin, definitions of olive oil products, uniform labeling and description of official instrumental and analytical methods for purity and quality assessment of EVOO [1].

Several research studies have reported the use of different analytical methods in the authentication of olive oil. Classical chromatographic methods such as liquid and gas chromatography have been used successfully in the detection of adulterants and authentication of olive oil [7,8]. Although these methods are effective in assuring the authenticity of EVOO, they are destructive, time-consuming, labor-intensive, and environmentally unfriendly. Therefore, there is a need to continuously develop alternative rapid methods to control and monitor fraud in the olive oil industry.

To date, some studies have reported the application of non-destructive techniques in combination with chemometrics to monitor fraud in olive oil. Vision technologies (i.e., hyperspectral imaging) and spectroscopic techniques (e.g., near-infra-red (NIR), UV-Vis, fluorescence, and Raman spectroscopy) are simple and rapid. They can therefore be used for online quality control as opposed to the conventional approaches. Fourier-transform infrared (FTIR) spectroscopy and multivariate data analysis have been used to detect adulteration in EVOO with some common edible oils [9,10]. Raman spectroscopy is another versatile sensitive vibrational spectroscopy method for detecting adulteration in olive oil. It has been used to detect common olive oil adulterants such as rapeseed, corn, soybean, and sunflower oils [5,11,12,13]. A few studies have also reported UV-Vis spectroscopy and multivariate data analysis for the detection and quantification of adulteration in olive oil with cheaper oils such as old olive, canola, sunflower, soybean, peanut, and sesame oils [14,15,16,17].

Hyperspectral imaging (HSI) has recently emerged as a powerful analytical technique for food quality and authenticity. HSI acquires both spatial and spectral data from an object. The spatial features of HSI allow the characterization of complex heterogeneous samples, while the spectral features provide comprehensive information on the chemical attributes of foods [18]. However, application of the HSI technique for the characterization of homogeneous liquid-based products such as edible oils has not been fully investigated [19]. A few studies have successfully applied HSI for the classification, authentication, and evaluation of the quality parameters of oil. HSI-NIR, in tandem with chemometrics, has been applied successfully in the discrimination of sesame oils [20,21]. HSI and different algorithms such as genetic algorithms (GA), least absolute shrinkage and selection operator (LASSO) and successive projection algorithm (SPA) were used to predict the peroxide value, acidity, and moisture content of olive oil [22]. A different study by [23] attempted to discriminate flavored olive oils based on HSI, but the findings were inconclusive. To the best of our knowledge, there is no information on the application of HSI in the detection of adulteration and authentication of EVOO. Furthermore, the efficacy of HSI technology in comparison with other rapid methods for the authentication of EVOO needs to be elucidated.

Spectroscopic and imaging techniques generate a considerable amount of information that cannot be used in its raw form to generate classification and regression models [24]. Chemometric techniques such as principal component analysis (PCA), linear discriminant analysis (LDA) and partial least squares (PLS) are used to build models for addressing food fraud [18].

This study investigated and compared HSI, FTIR, Raman, UV-Vis spectroscopies, and GC-MS techniques in tandem with chemometrics for (i) the discrimination of pure EVOO from non-olive oil; (ii) the detection of adulteration and the types of adulterants in olive oil; and (iii) the prediction/quantification of different adulterants (safflower, corn, soybean, canola, sunflower and sesame oils) in olive oil.

## 2. Materials and Methods

### 2.1. Samples

The pure extra virgin olive oil (EVOO), safflower oil, corn oil, soybean oil, canola oil, sunflower oil and sesame oil samples used in this study were purchased from a local retail store in Incheon, South Korea. All the samples were stored in the dark at 25 °C before analysis.

Fatty acid composition obtained by GC-MS was used to confirm the authenticity of the oil samples. Before adulteration, each edible oil sample was rapidly scanned using HSI-NIR (900–1700 nm). Principal component analysis (PCA) based on HSI spectra data was used to visualize clusters of EVOO and the adulterants. A random EVOO sample pooled from the EVOO cluster was then adulterated by homogeneously mixing with each adulterant in the range of commercial interest (0–20%) [25]. The levels of adulteration were 100:0, 99:1, 98:2, 96:4, 92:8, 88:12, 84:16 and 80:20 EVOO: adulterant, m/m). Sixty one samples, including 13 samples of EVOO, 6 vegetable oils and 42 adulterated samples, were prepared in triplicate for analysis. As such, a total of 183 (61 × 3) cases were analyzed for each technique. 

### 2.2. Chemicals, Reagents, and Standards

All the chemicals, the internal standard (Tridecanoic acid, C13:0) and the calibration standard (Supelco-37 FAME (fatty acid methyl esters) mix) used in the study were HPLC/GC grades from Sigma-Aldrich (St. Louis, MO, USA) and Merck (Darmstadt, Germany).

### 2.3. Fatty Acid Analysis by Gas Chromatography-Mass Spectrometry (GC-MS)

#### 2.3.1. Transesterification

The fatty acid composition of oils was determined by GC-MS according to [26] with modifications. Triacylglycerols (TAG) were converted to their corresponding fatty acid methyl esters (FAMEs) by trans-methylation. Approximately 20 mg of the oil sample was weighed in a test tube followed by the addition of 3.3 mL of methanol/hydrochloric acid (2 M in methanol) and vortexed for 10 s. Then, 50 µL of tridecanoic acid (C13:0) as the internal standard (10 mg/mL) was added to the samples and mixed for 30 s. The tubes with the samples were tightly sealed and incubated in a preheated oven at 90 °C for 2 h. The tubes were cooled to room temperature after being removed from the oven. Ultra-purified water (0.9 mL) was added to the tubes and vortexed for 20 s. Then, 2 mL of n-hexane was added and vortexed again for 30 s. The n-hexane layer containing the FAMEs was separated after centrifugation for 5 min at 4000 rpm. An aliquot of the n-hexane layer (100 μL) was mixed with 900 μL of n-hexane in a sample vial for GC-MS analysis.

#### 2.3.2. Preparation of FAME Standard Calibration Curves

Different concentrations of the fatty acid standard were prepared by diluting the Supelco-37 FAME mix stock standard (1 mL vial) with n-hexane. An aliquot of 5 µL of the internal standard (10 mg/mL) was added to each standard concentration in a vial insert. The final volume of each standard concentration was 200 µL. The concentration of the internal standard injected for GC analysis in both the sample and the standard was 12.5 µg/mL. The calibration curve for a specific FAME was constructed based on the ratio of the peak area of that FAME to the peak area of the internal standard.

#### 2.3.3. Data Acquisition on GC-MS

GC analyses were conducted with an Agilent 6890 GC (Agilent Technologies Inc., Santa Clara, CA, USA) with DB-23 capillary column (30 m × 0.25 mm ID, 0.25 μm film thickness) (Agilent Technologies Inc.). The capillary column was connected to an Agilent 5973 mass spectrometer (Agilent Technologies Inc.) with a quadruple analyzer and an electron energy of 70 eV. The analytical conditions were as follows. In the on-column inlet, the temperature was set to 50 °C, and the injection volume was 1 µL with a split ratio of 10:1. The column temperature was increased to 175 °C for 8 min and further increased to 235 °C at a gradient of 2 °C/min. Helium was used as the carrier gas at a flow rate of 0.7 mL/min. The temperatures of the detector and the injector were set to 230 °C and 250 °C, respectively. The mass spectra were recorded in the range of 30–800 *m*/*z* with a scanning frequency of 1.95 scans s^−1^.

#### 2.3.4. Identification and Quantification of FAMEs

FAMEs were identified by comparing their retention times versus those of pure standards (Supelco-37 FAME mix) analyzed under the same condition. Additionally, their mass spectra were further compared with the Mass Spectral Library of the National Institute of Standards and Technology (NIST). The total fatty acids (expressed as g/100 g) were quantified based on the calibration curves.

### 2.4. Spectral Acquisition by Hyperspectral Imaging

#### 2.4.1. Hyperspectral Imaging System and Software

A near-infrared hyperspectral imaging (NIR-HSI) system operating in the spectral range of 900–1700 nm was used for data acquisition. The system comprises a camera (Fx17e Specim), a light source that includes six tungsten halogen lamps (150 W) and an electric displacement system (40 × 20 Specim Lab Scanner) (Figure 1). The Lumo Scanner, Classic ENVI (IDL 8.7.2), and ENVI (version 5.5.2) software was used for image acquisition, normalization, and spectral data acquisition, respectively.

#### 2.4.2. Sample Image Acquisition

Each oil sample (10 g) was evenly distributed on a 60 mm diameter Petri dish. The camera’s exposure time, frame rate, speed, platform distance and the distance between the sample and lens and image resolution were optimized as 7.00 ms, 19.50 Hz, 2.6 mms^−1^, 400 mm, 15 cm, and 672 × 512 pixels, respectively. The sample was placed on the stage at a fixed position and the image was then captured by the NIR-HSI across the wavelength region of 900–1700 nm, with a spectral resolution of 4 nm. The hyperspectral images were stored in a three-dimensional form (x, y, λ), where x and y represent the spatial resolution and λ is the wavelength (nm). The images were acquired for every replicate of the sample resulting in three different hyperspectral images per sample. All measurements were done at room temperature (23 ± 2 °C). Due to the potential heating of samples by the halogen lamps, the HSI machine was switched on and off after every 20 analyses to allow for cooling. A fan was also used to circulate air continuously to maintain the room temperature. The raw image was corrected with black and white reference images based on Equation (1) to eliminate noise caused by the uneven dark current and uneven illumination distribution of the hyperspectral camera.
(1)$$\mathrm{Rc}=\frac{I-B}{W-B}\;$$
where Rc is the calibrated hyperspectral image, *I* is the raw hyperspectral image, *W* is the white reference image from the standard white calibration board with a reflectance of ~100% and *B* is the black reference image obtained by closing the lens of the camera (reflectance = 0%).

#### 2.4.3. Data Acquisition

To obtain the spectral data, an area of 50 × 50 pixels was selected as the region of interest (ROI), since it covered >80% of the sample’s area on the dish. The selection was done at the center of each sample to avoid interferences from the edges of the Petri dish [21,23]. ENVI software (version 5.5.2) was used to process the reflectance values of all pixels of ROI. The average reflectance spectra of all the pixels were used to calculate the mean reflectance spectrum for each sample of oil.

### 2.5. Spectral Data Acquisition by Fourier Transform Infrared Spectroscopy (FTIR)

The sample spectra in the mid-infrared region (400–4000 cm^−1^) were acquired with an FTIR spectrometer (Thermo Nicolet IS-5 spectrometer). The FTIR system was equipped with a horizontal attenuated total reflectance accessory (ZnSe ATR crystal) and optics detector (KBr) interfaced to a computer with the OMNIC software (Version 7.0 Thermo Nicolet). Briefly, 10 µL of the homogenized sample was pipetted and spread on a single bounce ZnSe ATR crystal. Each sample spectrum was scanned 32 times with a resolution of 4 cm^−1^ and a scan speed of 1 cm/s. The background air spectrum was collected before the acquisition of spectral data for every sample. The single-beam absorbance spectra for all the samples were collected and corrected against the background spectra. The ATR plate was cleaned by scrubbing with hexane twice, followed by acetone, and dried with soft tissue before subsequent analysis. The cleaned ATR crystal was checked spectrally to ensure that no residue from the previous sample was retained on the surface. The spectra were collected as absorbance values at each point.

### 2.6. Raman Spectroscopy Analysis

A Raman spectrometer (EZRaman-N 785 AIS) with a near-infrared laser operating at 785 nm as an exciting source was used for spectral measurements. The data were processed with EZ Raman Reader V8.4.9 software. The laser power and data acquisition time were set at 200 mW and 20 s, respectively, to achieve optimum Raman peak intensity [5]. A volume of 1 mL of each oil sample was pipetted into a 1.5 mL quartz vial and scanned by placing the sample in the sample holder and focusing the light onto the sample. The Raman spectrum of each sample was obtained by averaging five repeated scans, which was also the average spectrum of five operations by the software. The scanning resolution range for each test was 500 to 2000 cm^−1^ with a scanning interval of 1 cm^−1^. The background was collected by scanning an empty cuvette. Automatic baseline correction was applied to all the spectra to remove the luminescence background. The time required to complete the measurement of each sample was about 3 min. Toluene was used to calibrate the wavenumber and the intensity of the Raman instrument during setup.

### 2.7. Ultraviolet-Visible (UV-Vis) Spectra Acquisition

A computer-controlled spectrophotometer Jenway 7315 (Staffordshire, UK) equipped with a quartz cuvette of 1 cm optical path was used for spectral measurements. Jenway 73 series software was used for data acquisition and processing. Absorption spectra of the diluted oil samples (1:100) in isooctane (spectroscopy grade, Merck, Germany) were registered from 200 nm to 800 nm with 5 nm. The adjustment of the absorbance signal was performed using isooctane as the blank. The spectrum for each sample was obtained by averaging five repeated scans.

### 2.8. Chemometrics and Statistical Analysis: Model Construction

Before building classification and prediction models, the data from all the techniques were normalized and corrected for baseline shifts. As part of the data exploration process, principal component analysis (PCA) was first performed to visualize the data structures and separation of EVOO, edible oils and adulterated olive oils based on different analytical techniques using the Unscrambler X, CAMO Software AS (version 10.4, Oslo, Norway). The x-loadings from PCA analysis were used to link the chemical information related to the spectral profiles of oils and unsupervised classification by PCA.

Partial least squares (PLS) and PLS-discriminant analysis (PLS-DA) were used for dimension reduction of the spectral data and model construction. PLS is especially fit for usage in situations with more independent variables than samples. This is usually the case when dealing with spectral data, as in this study. PLS allowed the reduction of many independent variables (wavelengths of spectral data) to a much smaller number of latent variables. PLS was used to predict the concentration of adulterant oil in adulterated olive oil. PLS-DA is used in cases where the dependent variable is a categorical variable, such as the identity of oil used for adulteration in this study [27]. Two assessments needed to be made concerning the adulteration of olive oil. (i) PLS-DA classification: is the sample adulterated, and if so with which oil? and (ii) PLS: at what concentration m/m% is the adulterant added? To do this, a PLS-DA method was used to classify the samples (Figure 2). These classification results were used to feed the samples into PLS models (1 model per adulterant oil, 6 in total) for quantification of adulteration m/m%. All models were made based on a training set and validated on a testing set. The sample contents of these sets were varied; they were repeated ten times, and a randomly assigned (without replacement) 10-fold partition was made from the data at the onset of the analysis (Figure 2). A 10-times repetition was performed to (i) allow estimation of the number of latent variables in the PLS-DA and PLS models, which require estimation of the standard error of prediction (SE) and (ii) improve the prediction error estimation of adulterant classification and concentration.

Throughout the entire run (PLS-DA into PLS), a partition was held out as a testing set for validation, and as such never used to calibrate any of the models. This process was carried out for each partition, and for a 10 times repetition (on randomly assigned partitions) (Figure 2). In this way, the adulterant classification’s result impacted the adulterant concentration prediction because a wrong classification resulted in the allocation of a sample and its spectral data to the wrong model. Incorporating this error was necessary to maintain a realistic assessment of these models’ performance.

Selection of the number of latent variables was performed according to [27]. This procedure, called a “one standard error rule” chooses the best-performing model with the least number of latent variables, as such preventing overfitting of the data to the models [27]. Models were constructed with an increasing number of latent variables, based on the training data. The choice of the number of latent variables needs to be done based on the testing data. Testing data were put into the models and a percentage correct classification or root mean square error of prediction (RMSEP, which is RMSE of testing data) is computed for each model. Subsequently, the absolute best model (model *) i.e., the model with the lowest RMSEP or highest percentage CC, is identified. The final model (i.e., a robust reliable model) was selected as follows:

The final model is the model with the least number of latent variables that still satisfies one of the following conditions:(2)$$\left(\mathrm{for}\;\mathrm{PLS}-\mathrm{DA}\;\mathrm{models}\;\right)\;\mathrm{mean}\;\mathrm{CC}+\mathrm{SE}>\mathrm{mean}\;\mathrm{CC}\;\mathrm{of}\;\mathrm{model}$$
(3)$$\left(\mathrm{for}\;\mathrm{PLS}\;\mathrm{models}\right)\;\mathrm{mean}\;\mathrm{RMSEP}-\mathrm{SE}<\mathrm{mean}\;\mathrm{RMSEP}\;\mathrm{of}\;\mathrm{model}$$

In which mean CC is the mean of the CCs of each repetition (n = 10) and mean RMSEP is the mean of the RMSEPs of each repetition (n = 10). These mean CC and mean RMSEP, together with their SE, are used to compute the minimum number of latent variables to construct the final models for PLS-DA and PLS, respectively, as shown in Equations (2) and (3).

The performance of the models was evaluated by several performance parameters. The coefficient of determination (R^2^) was used to reveal the robustness of the models, R^2^_cv_ for cross-validation of the model (using the training set data) and R^2^_pred_. for validation of the model based on the test set [14,22]. The root mean square value of cross-validation (RMSECV) and root mean square error of prediction of the testing set (RMSEP) were used to quantify prediction errors. Additionally, residual predictive deviation (RPD) was also used to further assess the performance of the models. RPD is the ratio of the standard deviation of the measured dependent variable (adulterant’s concentration) values to RMSEP, which reveals the predictive ability of the model. RPD values lower than 2.0 are considered insufficient for prediction, while values 2.0 and 2.5 are considered for approximate quantitative predictions. On the other hand, values between 2.5 and 3.0 and higher than 3.0 are indicators of good and excellent predictions, respectively [28].

Models were constructed in RStudio version 1.4.1106. The partitions (10 times 10 random partitions of samples) were done with ‘createMultiFolds’ in the ‘hsdar’ package [29]. PLS-DA models were implemented via the ‘plsgenomics’ package [30]. PLS regression models were implemented via the package ‘pls’ [31].

## 3. Results and Discussion

### 3.1. Chemical Interpretation and PCA Unsupervised Classification

EVOO can be distinguished from other vegetable oils based on the composition of fatty acids (FA), degree of unsaturation and the presence of other specific minor components. The concentration of FA in EVOO and other edible oils used in the current study was determined by GC-MS (Table 1). The results for the main fatty acids in EVOO (palmitic, stearic, oleic, and linoleic acids) complied with the limits stipulated by the International Olive Council [1] and the Codex Alimentarius [32] and so did the other vegetable oils [33]. It is apparent that EVOO is high in MUFA (monounsaturated fatty acid) (C18:1, *cis*-9; 76%) (Figure 3a), while the other vegetable oils are predominated by polyunsaturated fatty acids, mainly C18:2 (Table 1).

As an initial step of data visualization, unsupervised classification by PCA was performed to gain insight into (1) the differences between the pure vegetable oils, and (2) the differences between pure EVOO and EVOO adulterated with safflower, corn, soybean, canola, sunflower, and sesame oils. PC1 and PC2 explained 98% of the total variation from GC-MS data. The variation in FA composition of oils allowed clear discrimination of EVOO from the other oils using GC-MS, as displayed by the PC plot (Figure 3b). From the correlation loadings (Appendix A), oleic and linoleic acids contributed to the structured variation in PC1 that separated EVOO from the edible oils. As per the present result, Ref. [7] have also demonstrated the variation in proportions of different fatty acids as important markers/fingerprints in discriminating EVOO from low-quality vegetable oils. Canola oil was positioned closer to the clusters of EVOO and adulterated olive oils. This could be explained by the high amount of oleic acid in canola oil compared with the other edible oils.

The HSI spectral reflectance curves of EVOO and other edible oils are shown in Figure 3c. The spectral profiles were similar for all the samples in the study. However, distinct absorption peaks were observed at wavelengths 1156, 1174, 1220, 1400, 1422, 1534 and 1580 nm, with EVOO dominating at wavelengths 1220, 1400 and 1422 nm. Additionally, reflectance peaks were evident at 1160 nm, 1181 nm, 1411 nm and 1658 nm, with EVOO dominant at 1160 nm and 1658 nm. These peaks are assigned to different functional groups such as C–H, C–C, C–N, C=O and O–H [21].

Exploring the HSI spectra data of all the samples with PCA showed that the first two principal components explained 94% of the total variance (Figure 3d). A few key wavelengths that contributed to most of the variance were identified from the loadings (Appendix A). The vibration modes of the functional groups at these wavelengths enabled the separation of EVOO from the edible oils. These were in the range of 1164–1174 nm, 1178–1195 nm, 1213–1234 nm, 1389–1400 nm, 1403–1421 nm and 1651–1680 nm. Similar to our study, Ref. [34] have attributed the spectral bands between regions 1100–1250 nm to second overtones of C–H stretching vibrations of the –CH_2_– and –CH_3_– groups of monounsaturated fatty acids and those assigned to their combinations. Additionally, the C-H vibrations are associated with fatty acids, PUFA (polyunsaturated fatty acid), in the range of 1100–1400 nm and 1600–1800 nm [20].

High absorbance by EVOO at wavelengths 1400, 1421, and 1658 nm can be explained by the vibration ranges of the C–H and O–H bonds of phenolic compounds previously reported to be between 1399–1699 by [35]. A clear separation was observed between EVOO and other edible oils, except for a few EVOO samples that clustered within the latter group. Interestingly, pure EVOO was clustered separately from adulterated olive oil on the PC plot. This striking distinctiveness suggests the potential use of the HSI technique in detecting the adulteration of EVOO with most of the edible oils used in this study.

Raman spectra for EVOO, safflower, corn, soybean, canola, sunflower and sesame oils are shown in Figure 4a. Raman shifts of interest were observed in the regions of 800 cm^−1^ to 1800 cm^−1^. The characteristic bands and their corresponding vibrations have been summarized in Table 2. Major differences in band intensities among oils are observed at 1081 cm^−1^, 1265 cm^−1^, 1300 cm^−1^, 1440 cm^−1^, 1655 cm^−1^ and 1745 cm^−1^.

From the PC scores plot in Figure 4b, PC1 and PC2 accounted for 89% of the total variance for Raman spectral data. Based on the PC loadings (Appendix A), vibrations at wavenumbers 1656 cm^−1^ (C=C stretching), 1263 cm^−1^ (=C–H bending), 1106 cm^−1^ (C–C stretching) and 970 cm^−1^ (C=C bending) contributed to the information in PC1 that distinctly separated the cheaper vegetable oils (adulterants) from EVOO [3]. Due to EVOO being low in polyunsaturated fatty acids, it exhibited low Raman intensities at these wavelengths. On the contrary, vegetable oils, being rich in linoleic acid, bear an extra C=C double bond that contributed to strong Raman shift vibrations at similar wavenumbers [36]. The difference in intensities separates the two classes of oils. Additionally, vibrations at 1293 cm^−1^ (C–H twisting), 1462 cm^−1^ (C–H bending), 1646 and 1660 cm^−1^ (C=C stretching) contributed to most of the variation in PC2 that separated some of the adulterated oils from the edible oils [12].

Figure 4c illustrates typical mid-infrared FTIR spectra of EVOO and other edible oils in the current study. The FTIR spectra of samples are dominated by peaks at 3010, 2927, 2857, 1747, 1463, 1373, 1241, 1164, 1095 and 721 cm^−1^. The dominant peaks in the regions of 2800–3100 cm^−1^, 1700–1800 cm^−1^, and 900–1400 cm^−1^ are due to C–H stretching, C=O stretching, C–O–C stretching and C–H bending, respectively [10,28]. The spectra of EVOO and other edible oils were similar by visual examination. However, differences were observed in peaks that were high in absorbance values. Vegetable oils displayed stronger absorption as compared to EVOO due to the presence of more polyunsaturated fatty acids.

The first two principal components cumulatively accounted for 91% of the total variance in the original FTIR spectra data (Figure 4d). Several key wavelengths and associated vibrations contributed to variations in PC scores based on the x loadings (Appendix A).

These wavelengths were 2927 and 2857 cm^−1^ (methylene absorbance peaks) due to asymmetrical and symmetrical stretching vibrations of aliphatic C–H in –CH_2_ and –CH_3_ groups; the single sharp peak at 1751 cm^−1^ is due to the ester carbonyl functional group of the triglycerides, a weak band at 1654 cm^−1^ is attributed to the stretching vibrations of the C=C group of *cis*-olefins, and finally, the bands around 1064 cm^−1^ and 809 cm^−1^ are associated with the stretching vibration of C=O ester groups and –CH_2_ wag. The vibrations at these wavenumbers were more pronounced in edible oils than in EVOO, contributing to their separation on the PC plot. The typical UV-Vis spectra for EVOO and the other edible oils used in the study are shown in Figure 4e. Maximum absorbance by EVOO was observed at 232 nm. Considering the unrefined form of EVOO, additional minor peaks were observed at around 410, 460 and 475 nm which corresponded to absorption by carotenoids [16,28]. Another minor peak at around 670 nm is associated with the presence of chlorophyll in EVOO [15].

PC 1 and PC 2 explain 88% of the variance in the UV-Vis spectral data of all the samples in the study, as depicted in Figure 4f. Based on the x loadings (Appendix A), important wavelengths were identified between 230–340 nm. The highest variation was contributed by absorption at 270 nm for PC 1, and at 250 nm and 285 nm for PC 2. Edible oils were characterized by more pronounced peaks between 230 nm to 290 nm due to the presence of conjugated dienes and trienes of unsaturated fatty acids. As mentioned previously, they are rich in linoleic and linolenic acids which are oxidized to isomerized conjugated dienoic and trienoic acids, respectively. However, EVOO is less susceptible to oxidation due to its low PUFA content and natural antioxidants. The conjugated dienes exhibit strong absorption at around 232 nm, while conjugated triene shows an absorption band at 270 nm in vegetable oils [37]. The differences in concentration and light absorption by these compounds may have partly contributed to the separation of edible oils from EVOO.

### 3.2. Discrimination of Oil Types and Determining Adulteration by PLS-DA

A PLS-DA model was built to classify pure EVOO and adulterated olive oil. Classification was performed by grouping the samples into 13 different classes: six adulterants, pure EVOO and olive oils adulterated with safflower, corn, soybean, canola, sunflower and sesame oils. This was performed to check the performance and capability of the model from each technique in distinguishing individual edible oils, authentic EVOO and adulterated olive oil from each other. Classification was also assessed based on whether olive oil was pure or adulterated, regardless of the type of adulterant. Classification could also be done by assigning each of the adulteration levels to a different class. However, it was more realistic to assign all the adulteration levels to a single class based on the type of adulterant. It is impossible to know the adulteration levels of external samples that are brought to the control laboratories for authentication. Additionally, the constructed models allow the detection of mixtures regardless of the adulteration percentages.

The optimum number of latent variables and the correct classification rates for all the techniques in the study are shown in Table 3. The best classification results on whether EVOO was authentic or adulterated were achieved by HSI spectra data and PLS-DA. HSI classified all the cases in the two groups correctly (100% CC_pred_). The other spectroscopic techniques achieved classification rates of >96%. Additionally, HSI-NIR and UV-Vis emerged as the best techniques for discrimination between distinct groups of either pure EVOO, edible oils and adulterated olive oils, with correct classification rates of 94% and 97% (CC_pred_), respectively.

As previously mentioned, it was important to evaluate how well the techniques separated the cheaper vegetable oils (adulterants) from each other. HSI, Raman, UV-Vis, and GC-MS classified all the cases correctly (100%), except for sesame oil (%CC = 90%). Each of the 10% of the sesame oil cases from these techniques were wrongly classified as EVOO + canola (HSI), corn oil (Raman and GC-MS) and safflower (UV-Vis technique). The FTIR and PLS-DA model classified 100% of corn and soybean oil cases correctly. Nonetheless, the misclassifications of edible oils by FTIR PLS-DA models were as follows: 33% of safflower and canola oil cases were falsely classified as corn oils, 23.3% of sunflower oil cases as either corn or canola oil, and 40% of sesame oil as sunflower oil.

Further PLS-DA classification of EVOO and different types of adulterated olive oils is shown in the confusion matrix (Table 4). The HSI discrimination model achieved a correct classification rate (CC_pred_) of 100% in classifying pure EVOO and olive oil adulterated with canola, soybean, and sesame oils. This means that not solitary case of adulterated olive oil was misclassified as authentic olive oil. Similarly, none of the EVOO samples were wrongly classified as adulterated olive oil by HSI. The lowest rate of correct classification of 75.2% was achieved from olive oil samples adulterated with sunflower oil. Falsely classified EVOO + sunflower samples were classified as either EVOO + canola or EVOO + sesame. There is a lack of published data on discrimination of EVOO from adulterated olive oil using the HSI technique. Our study, therefore, fills this knowledge gap. However, a recent study has reported successful discrimination of sesame oils using hyperspectral image analysis [20]; this was confirmed by this study, wherein no EVOO samples were classified as EVOO + sesame, nor vice versa, and an overall %CC_pred_ of 100% was achieved for both cases of EVOO + sesame (Table 4) and pure sesame oil.

PLS-DA discrimination by Raman spectroscopy classified 85.9% of pure EVOO cases correctly. The rest of the cases were falsely classified as either EVOO + safflower, EVOO + soybean or EVOO + corn. Additionally, only 0.5% of the cases were false negatives, as EVOO + safflower and EVOO + soybean samples were erroneously classified as pure EVOO. The model was, however, good at discriminating olive oils adulterated with canola and sesame oils, achieving a CC_pred_ of 98.6% and 100%, respectively. The Raman and PLS-DA classification results in our study closely corroborate those by [9]. In their study, 88% of olive/non-olive oils and 79% of adulterated olive oils were classified correctly.

The FTIR model was the second best at authenticating EVOO, with an average classification rate of 99%. Only 1% of pure EVOO cases were misclassified as EVOO + safflower (Table 4). Additionally, none of the adulterated olive oil cases were misclassified as authentic olive oil. The lowest rate of classification was observed from EVOO + sunflower samples (85.7%). Falsely classified EVOO + sunflower samples were classified as EVOO + sesame. However, an average correct classification of more than 88% was achieved by the models in discrimination of oils adulterated with safflower, corn, canola and sesame oils. Our results are closely compared to FTIR and PLS-DA findings by [9], who reported 100% average correct classification for olive oil samples. Our findings also reflect those of [10,38], who reported 100% correct classification of olive oils from other olive oil blends using FTIR and PLS-DA.

UV-Vis was the best model compared with the other techniques in the classification of different groups as authentic EVOO, adulterants, or specific adulterated olive oils. The model classified approximately 98% of pure EVOO correctly. Similar to HSI and FTIR techniques, false-negative cases were not detected, but 2% of the pure EVOO cases were falsely classified as EVOO + canola and EVOO + sunflower (Table 4). The model also classified 100% of the olive oil cases adulterated with safflower, canola, and sesame oils correctly. These results are consistent with a different study by [28] that reported a 100% successful discrimination rate of olive oil from olive oils adulterated with old olive oils using UV-Vis and orthogonal projections to latent structures discriminant analysis (OPLS-DA). The high classification rate from the UV-Vis and PLS-DA model is associated with the presence of different compounds, such as conjugated dienes and trienes, carotenoids, and chlorophyll derivatives and their varying concentrations, in pure EVOO and adulterated samples, which leads to different absorption capacities between 230–680 nm.

On the contrary, the PLS-DA classification model with GC-MS data was less effective in discriminating EVOO adulterated with safflower and corn oil compared with other technologies. This model, however, classified 93.3% of the EVOO cases correctly. The technique misclassified false-positive EVOO cases (6.6%) as either EVOO + corn or EVOO + sunflower (Table 4). The technique generated the highest rate of false negatives (6.1%) compared with spectroscopic methods. Our classification results closely compare with findings from [39]. In their study, GC-MS and PLS-LDA models allowed the detection of adulteration and the identification of the type of adulterant with prediction abilities above 90% and 85%, respectively. High classification accuracy based on PCA and PLS-DA models has been reported in triacylglycerol (TAG) data in the discrimination of EVOO’s geographical origin [8]. Contrarily, the same study reported lower prediction ability in models generated from FA composition. Unlike triglycerides, fatty acids provide insufficient information for accurate discrimination of authentic EVOO and adulterated olive oil, as shown in our study. In the case of spectroscopic techniques, they do yield indirectly through electromagnetic interaction additional molecular information that contributes to the discrimination of adulterated EVOO.

### 3.3. Prediction of Adulterant Concentration in Olive Oil by PLS

To answer the third question in our study, PLS regression was performed to quantify the concentration (m/m%) of different adulterants in contaminated olive oil. PLS calibration models were built and tested, as previously discussed in Section 2.8 and Figure 2. Similar to PLS-DA, PLS regression models were constructed to (1) predict adulterant concentration in olive oil regardless of the type of adulterant (overall models) and (2) predict adulterant concentration based on the distinct types of adulterants used in the study. The prediction performance of the models was evaluated by parameters such as regression coefficients (R^2^), error values (RMSEP) and residual predictive deviation (RPD). For a model to have high predictive ability, it must have a high R^2^ value, a high RPD and a low RMSEP. Our study assessed the performance of the prediction models in two ways, through (1) consideration of falsely classified samples from PLS-DA (PLS-DA classification errors), and (2) prediction with perfect knowledge of the identity of the adulterant (independent of PLS-LDA classification). The latter was performed to assess the quantitative performance of the PLS regression models themselves and to allow comparison with other studies. This approach was adopted because most studies fail to consider a classification step before building PLS regression models, or only focus on overall PLS models without consideration of individual adulterants. The statistical output parameters from PLS models for each approach are summarized in Table 5.

The best results for the overall HSI-PLS prediction model on the test set (considering PLS-DA misclassifications) were achieved with 17 LVs (latent variables) (Appendix A). The model displayed a high R^2^_pred_ (0.97), high RPD (5.9) and comparably low RMSEP (1.1%).

Similarly, the prediction models for quantification of different individual adulterants in olive oil were robust, as they are supported by high R^2^ values in the range of 0.95–0.98, high RPD values (4.4–8.5) and lower RMSEP values (0.9–1.6%). Test set models based only on the knowledge of the identity of the adulterant were also robust, with low RMSEP values (0.5–1.1%), improved R^2^_pred_ (0.97–1.00) and significantly higher RPD values (6.0–15.0). The Prediction and quantification of adulteration in EVOO with different vegetable oils based on HSI-NIR spectral data and PLS are reported for the first time in our study. The models were successful in predicting the levels of adulteration in olive oils mixed with safflower, corn, soybean, canola, sunflower and sesame oils. These findings provide further support for the hypothesis that HSI-NIR technology has the potential to quantify adulteration in EVOO involving cheaper vegetable oils and may be adopted by the food control and food regulatory bodies for the authentication of extra virgin olive oil.

The overall PLS regression model from Raman spectral data (based on PLS-DA dependent test set) was optimal using 8 LVs (Table 5). High R^2^_pred_ (0.92), high RPD (3.5) and a low RMSEP value (2.0%) for the overall test model confirm Raman as a rapid tool for detecting adulteration in olive oil. Our study, however, reports higher R^2^_pred_ and lower RMSEP values compared with similar study findings by [9]. Additionally, the PLS models were also good at predicting adulteration in EVOO with the specific adulterants used in the study.

On the one hand, the overall PLS regression model (considering PLS-DA classification errors) from UV-Vis spectra data was poor at predicting adulteration in olive oil. The test model had a low R^2^_pred_ (0.64), low RPD value (1.7) and a high RMSEP (4.1%). The same scenario was exhibited by the overall PLS regression model on test set of known identity (PLS-DA independent test set). The model displayed high variability, with levels of adulteration between 5 and 20% for all the types of adulterated olive oil (Appendix A). This was not the case for the other technique’s overall model, i.e., HSI, which sufficed for prediction (Appendix A). On the other hand, the models had a good predictive ability for olive oil blended with specific adulterants. The regression coefficients, error values and RPD for the PLS-DA dependent and PLS-DA independent test sets for the separate admixes of olive oil with safflower, corn, soybean, sunflower and canola ranged from 0.86–0.99 and 0.94–0.99, 0.8–2.5% and 0.8–1.7%, and 3.0–8.8 and 4.0–8.2, respectively.

Optimal results for the overall FTIR-PLS regression models on both PLS-DA dependent and independent test sets were achieved with 19 LVs. However, the models were not good enough for predicting adulteration in olive oil, as supported by the low RPD value (2.1). Although the R^2^_pred_ obtained from the overall model is not sufficiently good, the errors (RMSEP = 3.3%) are lower compared with the validation errors (15–17%) reported by [9]. Additionally, all the models for the quantification of individual adulterants’ concentration in olive oil considering PLS-DA classification errors were also poor in prediction (RPD = 1.1–1.7). Regardless of the known identity of each adulterant for the PLS-DA independent models, these models failed to predict the level of adulteration in EVOO + safflower, EVOO + corn, EVOO + soybean and EVOO + sesame oil blends. Contrary to our outcome, Ref. [10] have previously reported successful prediction of adulteration in olive oil adulterated with soybean oil. The error (RMSEP) and the R^2^_pred_ for their PLS model were 0.99 and 1.5% *v*/*v*, respectively. It is worth mentioning the consequences of misclassification on the quality of FTIR quantification models for EVOO + sunflower and EVOO + canola blended oils. However, the exclusion of errors from the falsely classified samples groups significantly improved the quality of these two models (Figure 5). This clearly depicts the consequences of wrong classification and the inherent transfer of errors on the quality of prediction models. This scenario elaborates on the importance of checking the quality of the model for the correctly classified samples (actual adulterant) versus the test set that may contain errors from PLS-DA misclassification. These models (adulterant-specific) are only recommended for practical quantification of an adulterant’s levels in EVOO provided the identity of the adulterant is known. Otherwise, they are deemed insufficient for cases in which PLS-DA precedes prediction by PLS regression due to the transfer of errors associated with misclassifications.

Lastly, PLS models from GC-MS data (fatty acids) were developed and tested as a conventional approach for predicting adulteration in olive oil. Regardless of the simplicity of the overall prediction model (6 LVs) on the PLS-DA dependent test set, it was not robust enough for the quantification of adulteration in olive oil. This was a rather remarkable outcome of the study. The R^2^_pred_ and RPD values for the model were low (0.53 and 1.4, respectively) and could therefore not be considered reliable for the accurate prediction of adulteration in EVOO. None of the individual prediction models on PLS-DA dependent test sets could also be able to quantify adulteration in olive oil with different adulterants. The R^2^_pred_, RMSEP and RPD values for the models ranged from 0.19–0.55, 5.3–7.6% and 0.6–1.3, respectively. Better results were, however, displayed by the overall prediction model on a test set of known identity. Even though the RMSEP of the model was significantly higher (7.5%) compared with the other techniques in the study, its RPD value (3.1) justifies its predictive capabilities. Despite the known identity of samples, separate models for each adulterant in the study failed at prediction as observed in the opposite group. As previously observed in some of the discussed techniques, the exclusion of erroneously classified samples significantly improved the parameters of prediction models. However, this was the only case for the overall GC-MS prediction model, not the other EVOO separate models with specific adulterants. It is apparent that the knowledge of sample identity does not influence the predictive abilities of the separate models.

## 4. Conclusions

Adulteration of extra virgin olive oil (EVOO) with cheaper edible oils is a rampant problem in the olive oil industry. The present study was designed to evaluate the potential of hyperspectral imaging and chemometrics in comparison with other spectral techniques or GC-MS for detecting fraud in EVOO. This study has been one of the first to investigate and report hyperspectral imaging and multivariate data analysis for the authentication and quantification of adulteration in EVOO. Hyperspectral imaging technology was able discriminate authentic EVOO from adulterated olive oil with 100% classification accuracy. HSI also emerged as the best technique in predicting the quantity of the adulterant in olive oil, as evidenced by the low RMSEP (1.1%), high RPD (6.0) and high R^2^_pred_ (0.97) of the overall prediction model. Based on our current findings, hyperspectral imaging shows potential as a reliable, fast, non-destructive, and environmentally friendly approach for the authentication and detection of adulteration in extra virgin olive oils, as opposed to lengthy and destructive chemical methods. However, further research will focus on emerging fraud in EVOO based on adulteration with other adulterant oils.

## Figures and Tables

**Figure 1 foods-12-00429-f001:**
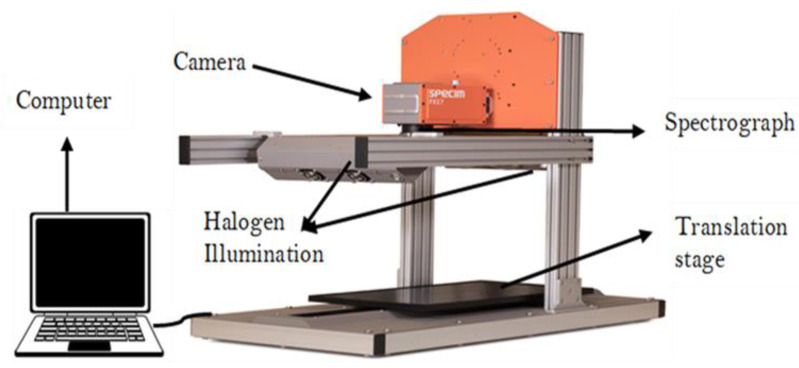
Near infrared (NIR) (900–1700 nm) hyperspectral imaging system.

**Figure 2 foods-12-00429-f002:**
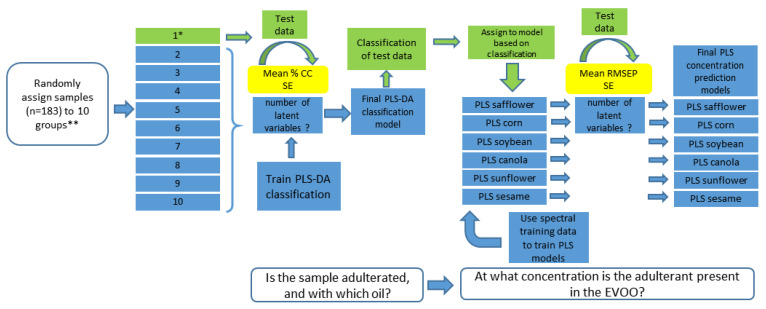
PLS model construction flow for the classification and quantification of the concentration of adulterants in EVOO; * the process for partition 1 is repeated for partitions 2 to 10 so that every partition is the test set once; ** randomly assigning samples is done 10 times (10 times 10 partitions are made); CC = correct classification.

**Figure 3 foods-12-00429-f003:**
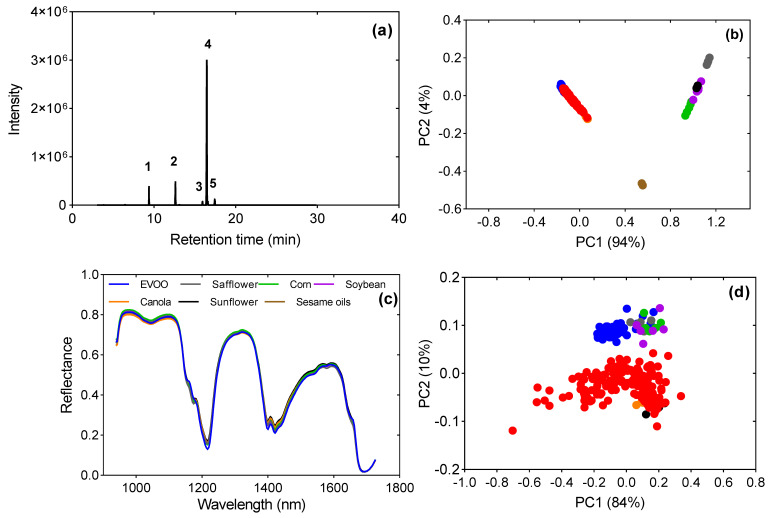
(**a**) Typical FAME GC-MS chromatogram for extra-virgin olive oil (EVOO). Peaks: 1—internal standard (tridecanoic acid); 2—C16:0; 3—C18:0; 4—C18:1, *cis*-9; 5—C18:2; (**b**) PC plot from GC-MS fatty acids composition data showing the relationship between EVOO, other edible oils and adulterated olive oil; (**c**) HSI spectra profile of EVOO and other edible oils; (**d**) HSI PC scores plot showing the relationship between EVOO, other vegetable oils and adulterated olive oil. Groups on the PC plots: ● EVOO; ● safflower oil; ● corn oil; ● soybean oil; ● canola oil; ● sunflower oil; ● sesame oil; ● adulterated olive oil.

**Figure 4 foods-12-00429-f004:**
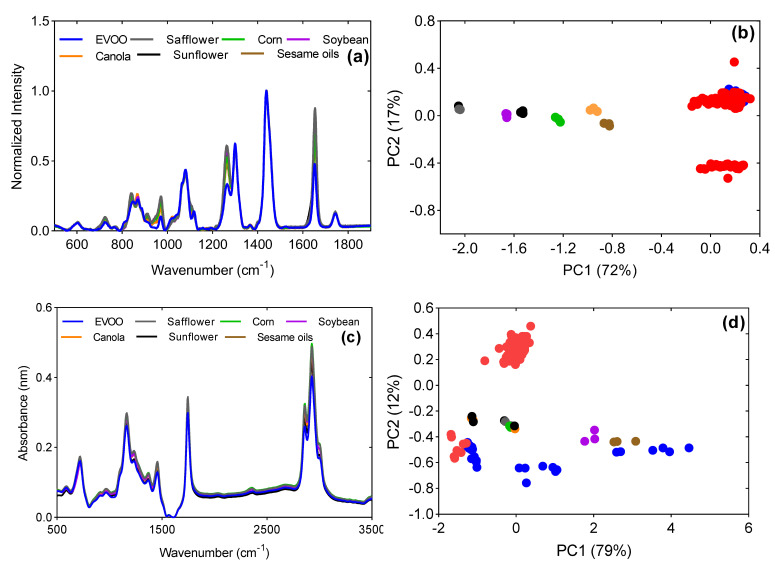
(**a**) Raman spectra of EVOO and other vegetable oils; (**b**) Raman PC plot showing the relationship between EVOO, other vegetable oils and adulterated olive oil; (**c**) Typical mid−infrared FTIR raw spectra for EVOO and other edible oils; (**d**) FTIR PC plot; (**e**) UV−Vis spectra of EVOO and other vegetable oils; (**f**) UV−Vis PC scores plot. Groups on the PC plots: ● EVOO; ● safflower oil; ● corn oil; ● soybean oil; ● canola oil; ● sunflower oil; ● sesame oil; ● adulterated olive oil.

**Figure 5 foods-12-00429-f005:**
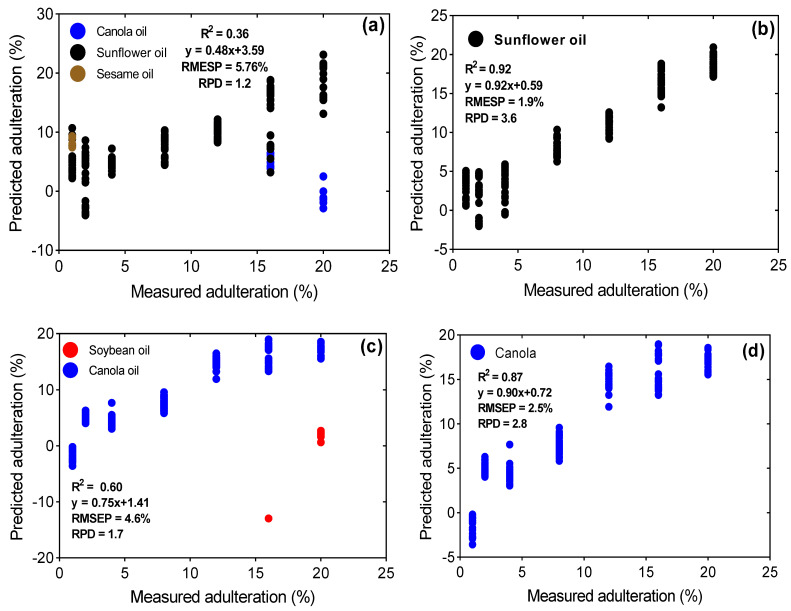
Consequences of PLS-DA misclassifications on the quality of PLS regression models: (**a**) FTIR and PLSR model on EVOO + sunflower test set, considering misclassifications from PLS-DA; (**b**) FTIR and PLSR model on EVOO + sunflower test set based on known identity of the test set; (**c**) FTIR and PLSR model on EVOO + canola test set, considering wrongly classified samples from PLS-DA; (**d**) FTIR and PLSR model on EVOO + canola test set based on the known identity of the test set.

**Table 1 foods-12-00429-t001:** Individual fatty acids and total SFA, MUFA and PUFA composition of EVOO and other edible oils as determined by GC-MS (%).

FAME	EVOO	Safflower Oil	Corn Oil	Soybean Oil	Canola Oil	Sunflower Oil	Sesame Oil
Palmitic (C16:0)	16.49 ± 0.49	8.99 ± 0.50	13.58 ± 0.81	12.53 ± 1.10	11.52 ± 0.40	8.88 ± 0.58	8.27 ± 1.22
Palmitoleic (C16:1)	0.11 ± 0.0	nd	nd	nd	0.06 ± 0.01	nd	nd
Stearic (C18:0)	4.17 ± 0.31	3.86 ± 0.42	2.36 ± 0.16	6.71 ± 0.41	2.40 ± 0.19	3.67 ± 0.47	5.51 ± 0.67
Oleic (C18:1 *cis*-9)	75.69 ± 0.73	10.38 ± 1.03	25.50 ± 1.47	17.76 ± 1.87	57.99 ± 0.58	20.18 ± 0.52	44.82 ± 0.65
Elaidic acid (C18:1 *trans*-9)	nd	0.13 ± 0.02	0.05 ± 0.02	0.46 ± 0.10	2.96 ± 0.21	1.01 ± 0.04	0.92 ± 0.26
Linoleic (C18:2)	3.48 ± 0.46	76.50 ± 1.88	58.30 ± 1.49	58.62 ± 1.56	17.57 ± 0.45	66.26 ± 0.44	40.49 ± 0.26
Linolenic (C18:3)	0.06 ± 0.02	0.14 ± 0.03	0.21 ± 0.03	3.93 ± 0.48	7.49 ± 0.53	nd	nd
SFA	20.66 ± 0.48	12.85 ± 0.90	15.94 ± 0.90	19.24 ± 1.16	13.92 ± 0.49	12.55 ± 0.41	13.78 ± 0.63
MUFA	75.80 ± 0.72	10.38 ± 1.03	25.50 ± 1.47	17.76 ± 1.87	58.05 ± 0.59	20.18 ± 0.44	44.82 ± 0.26
PUFA	3.54 ± 0.46	76.64 ± 1.89	58.52 ± 1.48	62.54 ± 1.80	25.06 ± 0.87	66.26 ± 0.44	40.49 ± 0.26

The values represent the mean percentage of total fatty acids ± SD; nd = not detected; EVOO = extra virgin olive oil; SFA = saturated fatty acid; MUFA = monounsaturated fatty acid; PUFA = polyunsaturated fatty acid.

**Table 2 foods-12-00429-t002:** Assignment of major Raman peaks in EVOO and other edible oils in the study.

Wavenumber (cm^−1^)	Molecule/Group	Vibrational Mode
868	–(CH_2_)_n_–	C–C stretching
968	trans RHC=CHR	C=C bending
1008	HC–CH_3_	CH_3_ bending
1150	–(CH_2_)_n_–	C–C stretching
1265	cis RHC=CHR	=C–H bending (scissoring)
1300	–(CH)_2_	C–H bending (twisting)
1440	–(CH)_2_	C–H bending (scissoring)
1525	RHC=CHR	C=C stretching
1650	cis RHC=CHR	C=C stretching
1750	RC=OOR	C=O stretching

Wavenumbers, functional groups and assigned vibrations are adapted from [3,12].

**Table 3 foods-12-00429-t003:** PLS-DA models of different techniques in the classification of edible oils, pure EVOO and adulterated olive oil (%CC_pred_. ^a^).

Technique	LVs	Classification as Pure EVOO, Adulterant (Type of Edible Oil) or Category of Adulterated EVOO * (Details in Table 4)	Classification as Pure EVOO or Adulterated Olive Oil
HSI	25	93.83 ± 0.34	100
Raman	14	87.76 ± 0.48	96.56 ± 0.16
UV-vis	26	96.72 ± 0.24	99.56 ± 0.18
FTIR	33	91.97 ± 0.47	99.78 ± 0.09
GC-MS	7	77.60 ± 0.32	93.72 ± 0.22

^a^ Average correct classification rate ± SE for prediction; LVs = latent variables. A total of 1830 cases were classified, originating from 61 oil samples in triplicate which were run through the PLS-DA for 10 repetitions (see Section 2.8 on ‘‘Chemometrics and Statistical Analysis: Model Construction” and Figure 2). EVOO = extra virgin olive oil; * EVOO adulterated with either safflower, corn, soybean, canola, sunflower or sesame oil.

**Table 4 foods-12-00429-t004:** PLS-DA classification (testing/validation data) of EVOO and EVOO adulterated with different edible oil adulterants.

		Predicted							
	Method	Class	EVOO	EVOO + Safflower	EVOO + Corn	EVOO + Soybean	EVOO + Canola	EVOO + Sunflower	EVOO + Sesame
**Measured**	HSI	EVOO	**390 (100%)**	0.0	0.0	0.0	0.0	0.0	0.0
		EVOO + safflower	0.0	**200 (95.2%)**	10 (4.8%)	0.0	0.0	0.0	0.0
		EVOO + corn	0.0	18 (8.6%)	**175 (83.3%)**	17 (8.1%)	0.0	0.0	0.0
		EVOO + soybean	0.0	6 (2.9%)	7 (3.3%)	**197 (93.8%)**	0.0	0.0	0.0
		EVOO + canola	0.0	0.0	0.0	0.0	**210 (100%)**	0.0	0.0
		EVOO + sunflower	0.0	0.0	0.0	0.0	45 (21.4%)	**158 (75.2%)**	7 (3.3%)
		EVOO + sesame	0.0	0.0	0.0	0.0	0.0	0.0	**210 (100%)**
	Raman	EVOO	**335 (85.9%)**	30 (7.7%)	17 (4.4%)	8 (2.1%)	0.0	0.0	0.0
		EVOO + safflower	6 (2.9%)	**180 (85.7%)**	4 (1.9%)	16 (7.6%)	0.0	0.0	4 (3.8%)
		EVOO + corn	0.0	0.0	**162 (77.1%)**	48 (22.9%)	0.0	0.0	0.0
		EVOO + soybean	1 (0.5%)	0.0	31 (14.8%)	**138 (65.7%)**	40 (19.0%)	0.0	0.0
		EVOO + canola	0.0	0.0	0.0	0.0	**207 (98.6%)**	3 (1.4%)	0.0
		EVOO + sunflower	0.0	6 (2.9%)	1 (0.5%)	0	6 (2.9%)	**197 (93.8%)**	0.0
		EVOO + sesame	0.0	0.0	0.0	0.0	0.0	0.0	**210 (100%)**
	UV-vis	EVOO	**382 (97.9%)**	0.0	0.0	0.0	6 (1.5%)	2 (0.5%)	0.0
		EVOO + safflower	0.0	**210 (100.0%)**	0.0	0.0	0.0	0.0	0.0
		EVOO + corn	0.0	0.0	**187 (89.0%)**	23 (11.0%)	0.0	0.0	0.0
		EVOO + soybean	0.0	3 (1.4%)	16 (7.6%)	**190 (90.5%)**	1 (0.5%)	0.0	0.0
		EVOO + canola	0.0	0.0	0.0	0.0	**210 (100%)**	0.0	0.0
		EVOO + sunflower	0.0	0.0	0.0	0.0	2 (1.0%)	**204 (97.1%)**	4 (1.9%)
		EVOO + sesame	0.0	0.0	0.0	0.0	0.0	0.0	**210 (100%)**
	FTIR	EVOO	**386 (99.0%)**	4 (1.0%)	0.0	0.0	0.0	0.0	0.0
		EVOO + safflower	0.0	**190 (90.5%)**	20 (9.5%)	0.0	0.0	0.0	0.0
		EVOO + corn	0.0	0.0	**194 (88.3%)**	16 (9.2%)	0.0	0.0	0.0
		EVOO + soybean	0.0	0.0	6 (2.9%)	**194 (92.4%)**	10 (4.8%)	0.0	0.0
		EVOO + canola	0.0	0.0	0.0	0.0	**193 (91.9%)**	17 (8.1%)	0.0
		EVOO + sunflower	0.0	0.0	0.0	0.0	0.0	**180 (85.7%)**	30 (14.3%)
		EVOO + sesame	0.0	0.0	0.0	0.0	0.0	5 (2.4%)	**205 (97.6%)**
	GC-MS	EVOO	**364 (93.3%)**	0.0	11 (2.8%)	0.0	0.0	15 (3.8%)	0.0
		EVOO + safflower	39 (18.6%)	**100 (47.6%)**	55 (26.2%)	0.0	0.0	16 (7.6%)	0.0
		EVOO + corn	0.0	59 (28.1%)	**110 (52.4%)**	1 (0.5%)	0.0	40 (19.0%)	0.0
		EVOO + soybean	0.0	10 (4.8%)	32 (15.2%)	**158 (75.2%)**	0.0	0.0	10 (4.8%)
		EVOO + canola	10 (4.8%)	20 (9.5%)	16 (7.6%)	0.0	**160 (76.2%)**	4 (1.9%)	0.0
		EVOO + sunflower	25 (11.9%)	0.0	13 (6.2%)	0.0	0.0	**162 (77.1%)**	10 (4.8%)
		EVOO + sesame	15 (7.1%)	5 (2.4%)	1 (0.5%)	0.0	0.0	0.0	**90 (84.2%)**

Diagonal figures in bold represent percentage correct classification. A total of 1830 cases were classified, originating from 61 oil samples in triplicate which were run through the PLS-DA for 10 repetitions (see Section 2.8 on ‘‘Chemometrics and Statistical Analysis: Model Construction” and Figure 2). EVOO = extra virgin olive oil.

**Table 5 foods-12-00429-t005:** Statistical parameters of PLS (partial least squares) regression for prediction of adulterant’s concentration (%) in olive oil using different techniques.

		Training Set (Cross-Validation)					Test Set (Considering PLS-DA Classification Errors) ^a^					Test Set (Known Adulterant Identity) ^b^				
Method	PLS-Regression Model	LVs	R^2^_CV_	Predicted vs. Measured Fitted Line	RMSECV (%)	RPD	LVs	R^2^_pred_	Predicted vs. Measured Fitted Line	RMSEP (%)	RPD	LVs	R^2^_pred_	Predicted vs. Measured Fitted Line	RMSEP (%)	RPD
HSI	Overall	19	0.97	y = 0.98x + 0.17	1.17	5.80	17	0.97	y = 0.98x + 0.21	1.14	5.91	18	0.97	y = 0.98 + 0.16	1.14	5.98
	EVOO + safflower	10	0.99	y = 0.98x + 0.18	0.53	13.14	9	0.95	y = 0.90x + 0.62	1.62	4.42	10	1.00	y = 0.98x + 0.22	0.48	14.23
	EVOO + corn	9	0.98	y = 0.98x + 0.11	0.87	8.06	9	0.98	y = 0.95x + 0.59	0.89	7.26	9	0.99	y = 0.99x + 0.38	0.76	8.90
	EVOO + soybean	8	0.98	y = 0.97x + 0.44	1.09	6.38	7	0.97	y = 1.00x − 0.03	1.14	6.00	7	0.98	y = 0.97x + 0.33	1.06	6.45
	EVOO + canola	7	0.99	y = 0.99x + 0.09	0.62	11.28	7	0.98	y = 0.99x + 0.23	0.86	7.80	6	1.00	y = 0.99x + 0.10	0.45	14.96
	EVOO + sunflower	4	0.99	y = 0.98x + 0.21	0.84	8.35	4	0.98	y = 0.97x + 0.24	0.91	7.10	4	0.98	y = 0.98x + 0.23	0.88	7.67
	EVOO + sesame	6	0.99	y = 0.98x + 0.17	0.69	10.13	5	0.98	y = 1.00x + 0.11	0.90	8.48	6	0.99	y = 0.98x + 0.20	0.57	12.00
Raman	Overall	8	0.92	y = 0.94x + 0.61	1.95	3.49	8	0.92	y = 0.94x + 0.62	1.96	3.48	8	0.92	y = 0.94x + 0.52	1.95	3.46
	EVOO + safflower	2	0.90	y = 0.90x + 0.93	2.17	3.21	2	0.86	y = 0.88x + 1.01	2.39	2.69	2	0.90	y = 0.94x + 0.40	2.17	3.12
	EVOO + corn	6	0.97	y = 0.95x + 0.51	1.16	6.03	2	0.95	y = 0.96x + 0.41	1.43	4.58	6	0.97	y = 0.94x + 0.52	1.20	5.65
	EVOO + soybean	5	0.98	y = 0.97x + 0.20	1.00	7.00	4	0.97	y = 0.93x + 0.46	1.29	5.23	5	0.98	y = 0.97x + 0.15	1.00	6.79
	EVOO + canola	3	0.96	y = 0.92x + 0.63	1.35	5.18	3	0.92	y = 1.09x − 0.03	2.55	2.86	3	0.97	y = 0.91x + 0.81	1.27	5.33
	EVOO + sunflower	3	0.93	y = 0.92x + 0.66	1.84	3.80	3	0.93	y = 0.98x + 0.09	1.78	3.76	3	0.93	y = 0.97x + 0.06	1.80	3.77
	EVOO + sesame	5	0.96	y = 0.92x + 0.61	1.39	5.03	5	0.96	y = 0.95x + 0.35	1.30	5.21	5	0.96	y = 0.94x + 0.40	1.29	5.27
UV-vis	Overall	8	0.63	y = 0.67x + 2.77	4.15	1.64	8	0.64	y = 0.65x + 3.13	4.06	1.66	8	0.64	y = 0.68x + 2.75	4.10	1.66
	EVOO + safflower	6	0.94	y = 0.96x + 0.11	1.76	3.97	6	0.94	y = 0.97x + 0.12	1.70	4.02	9	0.96	y = 0.97x + 0.06	1.45	4.69
	EVOO + corn	4	0.95	y = 0.99x + 0.19	1.67	4.18	2	0.90	y = 0.83x + 2.21	2.46	2.99	4	0.94	y = 0.97x + 0.35	1.69	4.00
	EVOO + soybean	9	0.98	y = 0.98x + 0.09	0.96	7.30	9	0.99	y = 0.99x	0.77	8.76	9	0.98	y = 0.98x + 0.10	0.89	7.83
	EVOO + canola	9	0.97	y = 1.02x − 0.05	1.19	5.91	8	0.93	y = 1.01x − 0.14	1.60	5.77	9	0.99	y = 1.01x + 0.02	0.84	8.11
	EVOO + sunflower	4	0.93	y = 0.92x + 0.44	1.91	3.66	4	0.94	y = 0.91x + 1.01	1.64	4.11	4	0.94	y = 0.90x + 1.07	1.65	4.14
	EVOO + sesame	4	0.98	y = 1.00x − 0.07	1.01	6.98	3	0.86	y = 0.91x + 0.63	2.40	3.39	5	0.98	y = 0.99x + 0.01	0.83	8.15
FTIR	Overall	16	0.69	y = 0.78x + 1.94	3.87	1.78	19	0.77	y = 0.85x + 1.41	3.26	2.07	19	0.77	y = 0.85x + 1.34	3.31	2.06
	EVOO + safflower	4	0.68	y = 0.74x + 2.00	3.98	1.76	5	0.62	y = 0.62x + 2.00	3.68	1.64	5	0.71	y = 0.80x + 1.99	3.79	1.81
	EVOO + corn	4	0.63	y = 0.74x + 2.27	4.34	1.62	2	0.17	y = 0.25x + 6.65	6.50	1.07	10	0.79	y = 0.93x + 0.53	3.32	2.04
	EVOO + soybean	2	0.53	y = 0.57x + 3.4	4.81	1.45	2	0.21	y = 0.29x + 5.56	6.19	1.11	2	0.45	y = 0.49x + 4.56	5.02	1.35
	EVOO + canola	8	0.84	y = 0.90x + 0.75	2.76	2.58	8	0.60	y = 0.75x + 1.41	4.59	1.73	8	0.87	y = 0.90x + 0.72	2.45	2.77
	EVOO + sunflower	10	0.91	y = 0.92x + 0.57	2.04	3.41	3	0.36	y = 0.48x + 3.59	5.76	1.21	10	0.92	y = 0.92x + 0.59	1.89	3.60
	EVOO + sesame	7	0.74	y = 0.87x + 1.18	3.72	1.89	4	0.43	y = 0.53x + 3.36	5.36	1.28	7	0.77	y = 0.84x + 1.61	3.26	2.08
GC-MS	Overall	6	0.90	y = 0.93x + 1.13	7.70	3.04	6	0.53	y = 0.68x + 3.45	4.85	1.38	6	0.90	y = 0.92x + 1.21	7.49	3.14
	EVOO + safflower	5	0.92	y = 0.96x + 1.31	11.52	3.70	2	0.22	y = 0.37x + 9.79	6.50	1.04	2	0.54	y = 0.63x + 6.09	5.41	1.25
	EVOO + corn	5	0.92	y = 0.96x + 1.08	11.44	3.69	4	0.27	y = 0.84x + 4.20	7.60	0.63	2	0.24	y = 0.43x + 7.30	6.77	0.99
	EVOO + soybean	5	0.89	y = 0.94x + 1.94	13.63	3.08	3	0.55	y = 1.02 + 2.44	6.33	0.99	3	0.60	y = 0.60x + 3.88	6.05	1.13
	EVOO + canola	3	0.88	y = 0.92x + 2.99	14.44	2.87	3	0.53	y = 0.95x + 2.52	6.24	1.05	2	0.29	y = 0.55x + 9.08	8.38	0.81
	EVOO + sunflower	4	0.91	y = 0.95x + 1.71	12.43	3.35	5	0.19	y = 0.53x + 3.81	6.86	0.82	2	0.24	y = 0.55x + 5.89	7.48	0.91
	EVOO + sesame	4	0.89	y = 0.94x + 2.01	13.65	3.09	3	0.51	y = 0.66x + 4.92	5.28	1.33	2	0.19	y = 0.30x + 7.79	6.45	1.04

^a^ Prediction based on all the test samples including falsely classified cases from PLS-DA (LDA dependent test set); ^b^ Prediction based on known identity of test samples (LDA independent test set); EVOO = extra virgin olive oil; RMSECV = root mean square of cross-validation; RMSEP = root mean square of prediction; RPD = residual prediction deviation; LVs = latent variables.

## Data Availability

Not applicable.

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
