# Peer review of "Hyperspectral Imaging and Chemometrics for Authentication of Extra Virgin Olive Oil: A Comparative Approach with FTIR, UV-VIS, Raman, and GC-MS"

_foods, 2023, doi:10.3390/foods12030429_

Round 1

Reviewer 1 Report

This manuscript describes a study on the using hyperspectral imaging and spectroscopic methods combined with chemometrics in the qualitative and quantitative assessment of the presence of adulteration in extra virgin olive oil.

The publication is written in a very careful way. This proves the authors' extensive knowledge of the methods of analytical chemistry and computational methods. It is very interesting to draw attention (in the introduction) to the risk of allergic reactions posed by adulterated EVOO.

My only remark concerns the lack of define and assessment of the applicability domain (AD) of the calculated models (according to OECD principles).

I also have a question for the authors: did they assess the usefulness of the models in assessing the adulteration of LVOO with two or more vegetable oils.

In my opinion, the work is very valuable and after minor revision can be processed further in Foods.

Author Response

My only remark concerns the lack of define and assessment of the applicability domain (AD) of the calculated models (according to OECD principles).

  • Standard errors of the mean were only used to check for classification and prediction accuracies. The range under which the models were are expected to make accurate predictions based on applicability domain were not defined.  An extension of this study will consider the applicability domain by reviewing data of the train set and test set models, as well as the conditions for the models in making predictions. 

I also have a question for the authors: did they assess the usefulness of the models in assessing the adulteration of LVOO with two or more vegetable oils.

  • The models were only assessed in two ways: i) classification as either authentic EVOO or adulterated olive oil; and quantification of adulteration regardless of the type of adulterant. Essentially, the samples were grouped into two as either pure olive oil or adulterated olive oil. ii) The performance of different PLS-DA and PLSR models was assessed on each specific adulterated olive oil i.e. PLSR models  EVOO+Safflower oil, EVOO+Corn oil, EVOO+Sesame oil PLSR models etc. The model was assessed for all the adulterants as a single group (overall model with samples adulterated with safflower, corn, canola, sesame, sunflower, and canola oils). 
  • However, if it is a question of mixing different adulterants with EVOO and assessing how our models performed on the same, then this was not assessed. This is a great suggestion worth considering in our next olive oil adulteration study.

Reviewer 2 Report

The manuscript use Hyperspectral imaging and chemometrics for authentication of extra virgin olive oil comparing different approaches.

I think the novelty is not enough and the application appears to be partially original.  However the author's workload is adequate. In general, the subject of the study reported in this paper is up-to-date and relevant in food science. Introduction describes the objectives of the study. Experiments were performed nicely and the techniques used were appropriated.

The work has been very well carried out and conclusions are sound and justified. Thus, in my opinion the manuscript can be processed further only after english revisions.

Author Response

English grammar has been improved and typographical errors in the document have been corrected.